# Repeat Induced Abortion among Chinese Women Seeking Abortion: Two Cross Sectional Studies

**DOI:** 10.3390/ijerph18094446

**Published:** 2021-04-22

**Authors:** Longmei Tang, Shangchun Wu, Dianwu Liu, Marleen Temmerman, Wei-Hong Zhang

**Affiliations:** 1School of Public Health, Hebei Medical University, Shijiazhuang 050017, China; 17600861@hebmu.edu.cn; 2Hebei Province Key Laboratory of Environment and Human Health, Shijiazhuang 050017, China; 3International Centre for Reproductive Health (ICRH), Ghent University, 9000 Ghent, Belgium; Marleen.Temmerman@UGent.be; 4National Research Institute for Family Planning, Beijing 100081, China; wushangchun@nrifp.org.cn; 5Centre of Excellence in Women and Child Health, Aga Khan University, Nairobi 00100, Kenya

**Keywords:** repeat induced abortion, family planning policy, Chinese women, cross-sectional study

## Abstract

Background: In China, there were about 9.76 million induced abortions in 2019, 50% of which were repeat abortions. Understanding the tendency of repeat induced abortion and identifying its related factors is needed to develop prevention strategies. Methods: Two hospital-based cross-sectional surveys were conducted from 2005–2007 and 2013–2016 in 24 and 90 hospitals, respectively. The survey included women who sought an induced abortion within 12 weeks of pregnancy. The proportion of repeat induced abortions by adjusting the covariates through propensity score matching was compared between the two surveys, and the zero-inflated negative binomial regression model was established to identify independent factors of repeat induced abortion. Results: Adjusting the age, occupation, education, marital status and number of children, the proportion of repeat induced abortions in the second survey was found to be low (60.28% vs. 11.11%), however the unadjusted proportion was high in the second survey (44.97% vs. 51.54%). The risk of repeat induced abortion was higher among married women and women with children [OR_adj_ and 95% CI: 0.31 (0.20, 0.49) and 0.08 (0.05, 0.13)]; the risk among service industry staff was higher when compared with unemployed women [OR_adj_ and 95% CI: 0.19 (0.07, 0.54)]; women with a lower education level were at a higher risk of a repeat induced abortion (OR_adj_ < 1). Compared with women under the age of 20, women in other higher age groups had a higher frequency of repeat induced abortions (IR_adj_: 1.78, 2.55, 3.27, 4.01, and 3.93, separately); the frequency of women with lower education levels was higher than those with a university or higher education level (IR_adj_ > 1); the repeat induced abortion frequency of married women was 0.93 (0.90, 0.98) when compared to the frequency of unmarried women, while the frequency of women with children was 1.17 (1.10, 1.25) of childless women; the induced abortion frequency of working women was about 60–95% with that of unemployed women. Conclusions: The repeat induced abortion proportion was lower than 10 years ago. Induced abortion seekers who were married, aged 20 to 30 years and with a lower education level were more likely to repeat induced abortions.

## 1. Introduction

Abortion is a global public health and social issue for women. It is estimated that 37 abortions occurred annually per 1000 women aged 15–44 years in the developing world in 2010–14 [1]. Induced abortion has been widely adopted as a family planning method for many years in China. According to the official statistics of China, the number of induced abortions in 2019 is about 9.76 million [2], and about 50% of these were repeated [3].

The family planning policy started in 1971 in China, and was divided into different stages in the past 49 years [4,5]. From 1970 to 1980, the limited birth policy was formed and comprehensively promoted [6]; between 1981 and 2000, the strict “one-child policy” was implemented [7]; from 2001, “one-child policy” began to adjust; from 2001 to 2012, the couples from single-child families could have two children, which was referred to as the “second child for couples both from single-child families policy” period; from 2013 to 2015, couples could have two children if one of them was a single child, this was referred to as the “second child for couples either from single-child family policy” period; finally, since the beginning of 2016, all couples could have two children, which is referred to as the “the universal two-child policy” period. Previous studies have analyzed the relationship between family planning policy and induced abortions. Through a descriptive analysis from 1979 to 2010, Wang considered that there was a relationship between family planning policy and induced abortion among married women [5]. Song analyzed the number of induced abortions from 2010 to 2013 in a hospital, and concluded that family planning policy has direct impact on the changes of induced abortions [8]. Zhao found that the number of induced abortions decreased from 2013 to 2017 in Kunshan, and indicated that the reason for the declination was due to changes in the family planning policy [9]. Based on these studies, it was assumed that repeated induced abortions were related to the changing of family planning policy. Lou observed 19,655 women seeking induced abortions in Xiamen from 2010 to 2013, and found that the proportion of repeat induced abortions has increased year by year [10]. Zhang et al. in Yangzhou in the years 2011 and 2012 observed that a total of 4242 adolescents sought abortion. Compared with the proportion of repeat abortions in 2011, it had increased in urban areas and decreased in rural areas in 2012 [11]. Chen et al. analyzed the data with regard to repeat induced abortion rates of one hospital in Guangdong province from 2008 to 2012, which showed a raise from 50.1% to 68.7% [12].These studies compared the proportion of repeat induced abortions among women seeking abortions in different years, but the study period of these studies is insufficient or study settings are limited. There is a lack of long-term comparison and extensive research to confirm our hypothesis.

There were multiple factors associated with repeat induced abortion. Firstly, social characteristics, such as age, education, migrants, parity, occupation, marriage, and fertility status showed a relation to repeat induced abortion [13,14,15,16,17,18,19]; secondly, personal behavior factors such as tobacco use, sexual debut, and number of sexual partners were also shown to be relevant [20,21,22,23,24,25]; thirdly, women who had experienced intimate partner violence or a family breakdown during childhood are at greater risk of repeat induced abortions [22,24,26,27]; finally, the practice of contraception and failure to use or incorrect use of contraceptives has increased the risk of repeat induced abortions [13,14,18,19,20,28]. Most of the studies used logistic regression to determine the characteristics of women with repeat induced abortions. To our knowledge, there is no study that focused on the differences between women who had induced abortion for the first-time and those who had repeat induced abortions by an appropriate model from quantitative and qualitative perspective synchronously.

Therefore, in this study, two nationwide cross-sectional surveys were used to compare the proportion of repeat induced abortions in two time periods of over 10 years, and we also used the zero-inflated negative binomial regression (ZINB) model to identify relevant factors of repeat induced abortion among women seeking induced abortion in China.

## 2. Methods

### 2.1. Study Design

Both cross-sectional surveys were parts of the EC-funded research projects, wherein one was carried out under the 6th Framework Program (PAFP: Post-Abortion Family Planning, FP-2002-INCODev-510961) from 2005–2007, and the other one was supported by the 7th Framework Program (INPAC: Integrating Post-Abortion Family Services into China’s existing abortion service in a hospital setting, FP7/2007-2013-282490) during 2012–2017.

### 2.2. Setting

The first survey was conducted in 24 hospitals in three cities of China (Beijing, Zhengzhou, and Shanghai), and the second one was conducted in 90 hospitals in 30 provinces of China (except the Tibet Autonomous Region).

### 2.3. Participants

During the survey, all women who sought an abortion within 12 weeks of pregnancy were invited to fill out the questionnaire before undergoing the abortion. They were informed that they had the freedom to refuse to participate in the survey. Women seeking abortion due to medical reasons or who were unwilling to participate in the survey were excluded. In the second survey, women who were below 18 years old were also excluded.

### 2.4. Variables

The questionnaire used in the first survey was developed on the basis of literature review, and the second survey was developed on the basis of the literature and the first survey questionnaire. Taking scientific and logical considerations into account, all questionnaires were discussed with team members which included the epidemiologist, clinician, statistician, and manger. The similar variables were collected in both the surveys, including the self-reported demographic and economic characteristics, history of induced abortions, causes of unintended pregnancy, and contraceptive practices. In this study, age was divided into 6 groups by every 5 years: ≤20, 21–25, 26–30, 31–35, 36–40, and ≥41 years old. Also according to the history of induced abortions, all women were divided into first induced abortion group (none) and repeat induced abortion group (greater than zero).

### 2.5. Data Collection

The data collection time was about 2 months in the first survey. According to the sample size estimation, the participants should have reached 200 in each hospital during the second survey period. In both surveys, a face-to-face survey method was used, and the data were collected by all participants by filling out the questionnaires through abortion service providers. Women under 18 years old could choose whether to complete the survey independently or while accompanied by their escorts.

### 2.6. Bias

To avoid selection bias, all women who met the study conditions during the survey were invited to participate in the survey. The characteristics of the surveyed population in the two surveys were different, which could cause confounding bias in our study. Therefore, propensity score matching (PSM) method and multiple regression were used in subsequent analysis to avoid confounding bias.

### 2.7. Sample Size Estimation

In the first survey, the sample size was directly obtained from the attached table in the book of “Determination of Sample Content in Hygiene Research” [29]. The repeat induced abortion proportion used was 50%, with a significance level set at 0.05, and an allowable error of 2.5%. The minimum sample size was 384 per city. As the sampling method used was cluster random sampling with an estimate effect of 2, the sample size was increased to twice its original size. Taking into account the failure rate and non-response rate, the sample size was finally decided to be 1000 per city. Altogether the estimated sample size of the first survey was 3000.

The second survey was the baseline for a cluster randomized controlled trial (RCT). So in the second survey, the sample size was estimated for cluster RCT by the Donner and Klar cluster randomization sampling estimation method [3]. The correlation coefficient between the clusters (hospitals) was 0.02, with a power of test of 80%, a significance level of 0.05, and the main outcome being a reduction from 3.5% to 1.5% through intervention. The minimum sample size was 100 per hospital. Taking into account the study as a 6-month intervention study and the withdrawal rate of 50%, it was finally determined that the participants included can be at least 200 per hospital. Altogether the estimated sample size of the second survey was 18,000.

### 2.8. Statistical Methods

The counts and proportions were used to describe categorical variables, and chi-square test was used to study the differences in proportion of repeat induced abortions and the related factors.

ZNBI was used to identify independent factors that were associated with repeat induced abortion. In ZNBI model, zero inflated procedure was used to analyze the independent factors for the proceedings of repeat induced abortion or not, while count procedure was used to find the frequency of factors associated with repeat induced abortions.

Covariates, including age, occupation, education, marital status, and number of children were matched in the PSM method. The match tolerance was 0.1, and match ratio was 1.

All analysis were performed by SPSS 24.0 (IBM Corp., Armonk, NY, USA) or R pscl package. A *p*-value of <0.05 on two sides was considered to be statistically significant.

### 2.9. Ethics

Both surveys were first approved by the Ethical Committee of the National Research Institute for Family Planning (China) and followed by the Ethical Committee of Ghent University (Belgium). All participants in the survey signed the informed consent form before survey. In the first survey, the informed consents of women under 18 years old were signed by their parents or sexual partner. The questionnaire was anonymous and data protection complied with the European Union (EC) regulations on date protection and privacy guidelines.

## 3. Results

### 3.1. Sociodemographic Characteristics of Women Seeking Induced Abortion

Altogether, 9554 and 18,503 women participated in two surveys separately, and after excluding missing values and logical errors, 26,125 women (9497 in the first survey and 16,628 in second survey) were included in this study. The median age of participants in the two surveys was 23 years (range: 15–49) and 27 (range: 18–49).

In the first survey, about 70% of women who sought an induced abortion under the age of 25, while in the second survey, this proportion was 38.77%. The age of women was statistically different between the two survey periods (*p* < 0.001). The education level of women seeking induced abortion in the first survey was significantly higher than those in the second survey (*p* < 0.001). The proportion of unemployed women and managers remained low, and the proportion of students and service industry staff remained higher in the first survey. There was a significant difference between the two surveys (*p* < 0.001). The proportion of unmarried and childless women in the first survey was also significantly higher than that included in the second survey (*p* < 0.001). (Shown in Table 1).

### 3.2. History of Induced Abortion among Women Seeking Abortion

Among women seeking abortion, 48.55% reported having at least one previous induced abortion. The repeat induced abortion in the two studies were 44.98% and 50.02%, respectively (shown in Table 2). By adjusting for age, occupation, education, marital status and children, a total of 6732 women (3366 women during each survey period) were successfully matched, and the sociodemographic characteristics of women seeking induced abortion were exactly the same between two survey periods. Through PSM control, the proportion of repeat induced abortions significantly declined in the second survey (shown in Table 2).

The highest induced abortion frequency was 8 in the first survey, and 11 in the second survey, with an average frequency of 1.53 and 1.60, respectively. While under PSM, repeat induced abortion frequency in the second survey was lower than that in the first survey (Shown in Table 3).

### 3.3. Associated Factors to Repeat Induced Abortion

Women with a lower education level were at higher risk of repeat induced abortion [Compared with ‘University or above’ group, OR_adj_(95% CI) for ‘junior middle school’ was 0.12 (0.05, 0.30), for ‘senior middle/technical school’ was 0.19 (0.10, 0.36), for ‘college’ was 0.46 (0.31, 0.69)]. Compared with unemployed women, the repeat induced abortion risk of service industry staff was higher [OR_adj_(95% CI): 0.19 (0.07, 0.54)]. The risk of repeat induced abortion was higher for those who were married and already had children [OR_adj_(95% CI): 0.31 (0.20, 0.49) and 0.08 (0.05, 0.13)] (Shown in Table 4).

Compared with women under the age 20, the average frequencies of induced abortions was increased with the age [IR_adj_(95% CI)for ‘21–25’ was 1.78 (1.62, 1.95), for ‘26–30’ was 2.55 (2.31, 2.81), for ‘31–35’ was 3.27 (2.96, 3.62), for ‘36–40’ was 4.01 (3.62, 4.45), for ‘more than 41’ was 3.93 (3.47, 4.46)]. The frequency of induced abortions of women with other lower education levels were all higher than those with university or above education [IR_adj_(95% CI) for ‘college’ was 1.18 (1.11, 1.25), for ‘senior middle/technical school’ was 1.30 (1.23, 1.38), for ‘junior middle school’ was 1.36 (1.27, 1.44), and for ‘primary school or below’ was 1.29 (1.15, 1.45)]. Except for managers, the frequency of induced abortions in women of other occupations was about 60–95% when compared to unemployed women [IR_adj_(95% CI) for ‘company staff’ was 0.90 (0.86, 0.95), for ‘service industry staff’ was 0.95 (0.91, 0.99), for ‘student’ was 0.62 (0.52, 0.75), for ‘technical staff’ was 0.84 (0.77, 0.92), for ‘other’ was 0.82 (0.75, 0.90)]. The frequency of repeat induced abortions in married women was 0.93 (95% CI: 0.90–0.98) of unmarried women, while the frequency for women with children was 1.17 (95% CI: 1.10–1.25) compared to for women without children (Shown in Table 4).

## 4. Discussion

### 4.1. Findings and Interpretation

In our study, the repeat induced abortion proportion was about 50% in the second survey. In recent years, half of the induced abortions were repeat induced abortions and the result was also similar to other surveys conducted in China [30,31,32]. Compared with countries where abortions are illegal [33,34,35], as in Asian countries [36,37], China had a higher proportion of repeated induced abortions. But it was basically same as in European and American countries [38,39]. Firstly, the reason for the similar proportion of repeat induced abortions with European and American countries might be due to the release of the family planning policy in China. The second reason was that the characteristics of the surveyed population were different, which was not comparable between different countries.

The characteristics of survey population in two surveys were different. Although the proportion of repeat induced abortions in the second survey was high, after having adjusted the covariates by the PSM method, the proportion of repeat induced abortions in the second survey was significantly lower than that in the first survey. The first reason for the declination in repeat induced abortions was the adjustment of China’s family planning in China. Both the surveys were conducted at different stages of family planning policies. With the opening of the family planning policy, the induced abortions related to family planning policy was decreased, and accordingly repeated induced abortions were decreased. Another reason was the provision of family planning services. In 1994, the government of China has emphasized the ‘people-oriented’ service concept and actively carried out contraceptive counseling services actively [40]. Especially since 2011, China has started to implement the Post Abortion Care (PAC) project to provided scientific contraceptive counseling for women seeking induced abortion. Studies had shown that PAC services could improve contraceptive effectiveness effectively, reduce the risk of unintended pregnancy, and repeat induced abortion [41,42,43,44,45,46].

The risk of repeat induced abortion was different among women with different social-demographic characteristics who were seeking induced abortion.

Women who were married and with at least one child were more likely had repeat induced abortions. Influenced by Chinese traditional moral concepts, the sexual intercourse rate among unmarried young women was about 20% in China [47,48,49]. The less sexual intercourse, the lower the chance of repeat induced abortions would be. In China, the education expenditure was the fastest growing category in household consumption expenditure. It was estimated that the average annual education expenditure of urban households would be about 9000 yuan, with an average annual growth rate of 20% [50]. So, most of the families would rather have only one child to ensure the quality of his/her education [51]. A survey on Chinese fertility showed that the number of Chinese ideal children has shown a downward trend since the 1980s, and now the average number of ideal children is basically stable between 1.6 and 1.8 [52].

After having one or two children, women became reluctant to become pregnant. Therefore, married women had a higher risk of repeat induced abortion than unmarried women, and the frequency of induced abortion after giving birth were increased by 17%.

The frequency of repeat induced abortions increased with age. It is biologically rational that older women were at greater risk of unintended pregnancy and induced abortion [14,17]. Moreover, the relationship between age and repeat induced abortions might be influenced by the expected number of children. Aged women are often those who have a sufficient number of children, so they undergo an abortion in the case of an unintended pregnancy [53].

As discussed in the previous studies, women with a high level of education had a lower risk of repeat induced abortions, in which the frequency decreased with an increased level of education. Education level directly affected the acceptance of contraceptive knowledge. Women with higher education level had stronger awareness and ability of contraceptives, thus reducing the number of repeat induced abortions [30].

In all occupations, service workers were at higher risk of repeated induced abortions. Most of the service staff were young immigrant women with lower education, who had little access to family planning services [54]. The demand for reproductive health services remained high, while the utilization rate remained low, which can increase the risk of repeat induced abortions.

### 4.2. Strengths and Weaknesses

As far as we know, this was the first time in a decade that China had conducted an extensive survey to analyze repeat induced abortions. Both surveys were international multi-center collaborative studies led by the same team. In particular, the second cross-sectional survey was conducted nationwide by including 30 of 31 provinces in mainland China (except for Tibet).

The first limitation of our study was relying on self-reporting of sensitive items, such as the frequency of induced abortions. Although anonymous self-report questionnaires were used, they might be vulnerable to social expectations bias. Secondly, our study was based on the data before the implementation of the ‘comprehensive two-child policy’, and so the changes in the level of repeat induced abortions during ‘comprehensive two-child policy’ required further study. Finally, this study involved only personal socio characteristics, and not sex-related factors, which might in turn affect the results.

## 5. Conclusions

In recent years, the proportion of repeat induced abortions was about 50%, which was lower than 10 years ago. Induced abortion seekers who were married, aged 20 to 30 years, and with lower education levels were more likely undergoing repeat induced abortions. Interventions that aim at reducing repeat induced abortions should be strengthened and targeted to high-risk groups.

## Figures and Tables

**Table 1 ijerph-18-04446-t001:** Comparison of sociodemographic characteristics of women seeking induced abortion between two surveys.

	First Survey (2005–2007) (*n* = 9497)		Second Survey (2013–2016) (*n* = 16,628)		*p*
	*n*	%	*n*	%	
Age					<0.001
≤20	1551	16.33	1404	8.44	
21–25	5042	53.09	5043	30.33	
26–30	1522	16.03	4960	29.83	
31–35	842	8.87	3209	19.3	
36–40	428	4.51	1628	9.79	
≥41	112	1.18	384	2.31	
Education					<0.001
Primary school or below	2332	24.56	4609	27.72	
junior middle school	2415	25.43	4348	26.15	
senior middle school/technical school	2999	31.58	4381	26.35	
College	1596	16.81	3024	18.19	
University or above	155	1.63	266	1.60	
Occupation					<0.001
Unemployed	1435	15.10	4058	24.4	
Company staff	3764	39.60	5639	33.91	
Service industry staff	2360	24.80	3182	19.14	
Student	832	8.80	771	4.64	
Manager	360	3.80	852	5.12	
Technical staff	608	6.40	1010	6.07	
Other	138	1.45	1116	6.71	
Marital status					<0.001
Unmarried	5030	52.96	5039	30.30	
Married	4467	47.04	11,589	69.70	
Children					<0.001
None	7075	74.50	1146	6.89	
More than 1	2422	25.50	15,482	93.11	

**Table 2 ijerph-18-04446-t002:** History of induced abortion among women seeking induced abortion in two surveys.

	Raw Data (*n* = 27,124)				PSM Data (*n* = 6732)			
	First (2005–2007)		Second (2013–2016)		First (2005–2007)		Second (2013–2016)	
	*n*	%	*n*	%	*n*	%	*n*	%
none	5226	55.03	8058	48.46	1337	39.72	2992	88.89
≥1	4271	44.97	8570	51.54	2029	60.28	374	11.11
*p*-value	<0.001				<0.001			

Note: Chi-square test was used for comparing the data between the two surveys. PSM was matched according to age, occupation, education, marital status, and number of children.

**Table 3 ijerph-18-04446-t003:** Number of previous induced abortion among participating women in two surveys.

	Raw Data				PSM Data			
	First (2005–2007)		Second (2013–2016)		First (2005–2007)		Second (2013–2016)	
	*n*	%	*n*	%	*n*	%	*n*	%
1	2786	65.23	5217	60.88	1149	56.63	282	76.47
2	966	22.67	2164	25.25	528	26.02	53	14.17
3	348	8.15	749	8.74	232	11.43	15	4.01
4	117	2.74	286	3.34	82	4.04	18	4.81
5	38	0.89	88	1.03	26	1.28	1	0.27
>=6	16	0.37	66	0.77	12	0.59	1	0.27
Average frequency	1.53		1.60		1.70		1.40	
*p*-value	<0.001				<0.001			

Note: Chi-square test was used for comparison between the two surveys.

**Table 4 ijerph-18-04446-t004:** Factors associated with repeat induced abortion in multivariable analysis.

	Zero Model *	Count Model ^#^
	OR_adj_(95% CI)	IR_adj_(95% CI)
Age		
≤20	1	1
21–25	1.46 (0.69, 3.10)	1.78 (1.62, 1.95)
26–30	1.27 (0.56, 2.90)	2.55 (2.31, 2.81)
31–35	0.44 (0.15, 1.28)	3.27 (2.96, 3.62)
36–40	0.15 (0.02, 1.19)	4.01 (3.62, 4.45)
≥41	0.24 (0.02, 3.66)	3.93 (3.47, 4.46)
Education		
Primary school or below	0.69 (0.22, 2.22)	1.29 (1.15, 1.45)
junior middle school	0.12 (0.05, 0.30)	1.36 (1.27, 1.44)
senior middle school/technical school	0.19 (0.10, 0.36)	1.30 (1.23, 1.38)
College	0.46 (0.31, 0.70)	1.18 (1.11, 1.25)
University or above	1	1
Occupation		
Unemployed	1	1
Company staff	0.73 (0.47, 1.16)	0.90 (0.86, 0.95)
Service industry staff	0.19 (0.07, 0.54)	0.95 (0.91, 0.99)
Student	1.09 (0.52, 2.29)	0.62 (0.52, 0.75)
Manager	1.16 (0.61, 2.21)	1.02 (0.93, 1.11)
Technical staff	0.99 (0.50, 2.00)	0.84 (0.77, 0.92)
Other	0.79 (0.35, 1.82)	0.82 (0.75, 0.90)
Marital status		
Unmarried	1	1
Married	0.31 (0.20, 0.49)	0.93 (0.88, 0.98)
Children		
None	1	1
More than 1	0.08 (0.05, 0.13)	1.17 (1.10, 1.25)
Survey		
First	1	1
Second	13.93 (8.40,23.12)	0.92 (0.88, 0.96)

Note: *: Unrepeated induced abortion was the response probability in zero process of the ZINB model. OR_adj_ > 1 indicated more likely occurrence of “false zero” situation, or repeat induced abortion risk was lower than that of control group, OR_adj_ < 1 indicated the risk was higher than that of control group. ^#^: Repeat induced abortion frequency was analyzed in count process of ZINB model, IR_adj_ > 1 indicated that the average frequency of repeat induced abortion was several times that of control group, while IR_adj_ < 1 indicated that the average frequency of repeat induced abortion was several percentage that of control group.

## Data Availability

The data presented in this study are available on request from the corresponding author. The data are not publicly available due to the privacy restrictions.

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
