# Peer review of "Repeat Induced Abortion among Chinese Women Seeking Abortion: Two Cross Sectional Studies"

_ijerph, 2021, doi:10.3390/ijerph18094446_

Round 1
Reviewer 1 Report
I congratulate authors for the improvements made in the paper.
Reviewer 2 Report
The manuscript has been improved in all its parts as suggested and may be accepted for publication in the present form.
This manuscript is a resubmission of an earlier submission. The following is a list of the peer review reports and author responses from that submission.
Round 1
Reviewer 1 Report
Dear authors,
This paper address a very interesting and challenging topic, such as repeated induced abortion in China.
Overall, the paper is well written and the results are described in a notable way. The weakness of the paper that I have detected are in the introduction and discussion section.
I address some recommendations for authors in order to improve the quality of the study:
Introduction: this section is very concise, and there is a lack of scientific background of the topic. Taking into account that the study is conducted in China, this section should be better contextualized. I noted that in the conclussion section the authors mention different laws about Chinese family planning that probably would be described in the introduction. The authors state that the rates of abortion in China are similar to European countries but they don’t explain in what sense these rates are similar. Please, say more about this.
I think there is a mistake at the beginning of the introduction, second line: women aged 15-14 years in the developing world…? Please, check it.
Aims. The aim is to analyze the stability or change in repeat induced abortion rates in China or the factors associated to repeat induced abortion? Or both? Please, describe this section in depth. Moreover, there is not clear hypothesis and there is not scientific literature supporting the study. Is it an exploratory study in nature? Please, say more about that.
Methodology: Sample: The sample is noteworthy in both studies but there is no clear for me how the authors recruited the sample. How many women refused to participate? How did you proceed with those women under 18 years old. Did they parents agree to participate? Did they completed the questionnaire during the visits? After or before abortion? I have noted that in the first cross-sectional study women were recruited from clinical and from hospitals in the second study. Should it be a factor to consider?
Measures: Measure is not well described. It is an ad-hoc measure? It is based on national or international studies/guidelines?
Results: There is a mistake in line 126. Age appears twice. Please correct.
Results regarding factors related to repeat induced abortion. Please, describe results following the variables order as represented in the table, or viceversa.
The survey factor presents a very wide CI in the zero model.
Conclusion: this section address some aspects don’t mentioned in the introduction section. Again, the lack of clarity in the aims doesn’t permit to understand the main aim of the study (induced abortion of repeat induced abortion or both). I was wondering if the authors can compare these results with other studies carried out in other countries and if some policy implications must be addressed. In overall, if we don't understand the Chinese context, it is difficult to understand the results.
Reviewer 2 Report
Two surveys about 10 years apart are presented.
Both were funded by different EU famework programmes. This should be mentioned under funding and grant numbers given.
There is something wrong with Table 2. Wjhat does one and >1 mean in the left hand column. If it means no of previous abortions hove can there be 4,271 having a first abortion who are also have >1.
If it means number of living children this should be made clear.
Minor points
- put the years of each survey above each column in the results tables
- Two significant fiures is sufficient for percentages, odds ratios and confidence intervals.
- Spell check the whole paper
- Needs english language grammar check.
Suggest the autors ask Marleen Temmerman to give it a final careful read and edit.
Reviewer 3 Report
This study addresses an important and little studied issue, repeat abortion in China.
Introduction (1.) and Literature Review (none)
The problem statement needs much more development and the authors need to include a literature review. Findings from the study should not be presented in the Introduction. Instead, the authors should (1) explain the topic of the manuscript and why it is important and (2) review the extant literature including the hypotheses and variables that this literature suggests.
Methods
2.1 and 2.2 Study Design and Setting
The authors need to acknowledge and explain that both the sample sizes and the context (e.g., clinics vs. hospitals differ significantly so the design of this study is not balanced and why that is important.
2.3 Participants
The authors need to explain how study participants or respondents were “invited” and how free they were to decline to participate in the study.
2.4 Variables
The lack of a literature review means that the selection of independent variables seems to be subjective or come out of nowhere.
2.5 Data Collection
This section needs more discussion about the survey instrument as well as how the questionnaire was administered.
2.6 Bias
Needs explanation rather than just reassurance.
2.7 Sample size estimation
Need to explain why this is necessary and how this balances the two time periods.
2.8 Statistical methods
No suggestions, except to explain PSM.
2.10 Ethics
Similar comments as in section 2.6
Discussion
This section should not introduce new material. That belongs in the literature review.
Need to separate out findings from what the data only suggest (e.g., effect of enhanced family planning in China.
Reviewer 4 Report
Herein the authors aimed at determining the factors
associated with the repeated abortion among women undergoing induced abortion in China.
The study should be improved in design, methods and result presentation. In addition, extensive editing of English language and style is also required.
Here some spelling errors.
1) line 84: sample and not simple
2) line 113: different and not difference
3) line 116: the proportion of unemployed/unemployed women and manager was low....this sentence is not clear
4) line 117: higher and not high
5) line 126: age is repeated
6) line 139: higher and not high
7) line 140: children and not child
